# A Large Scale Sample Complexity Analysis of Neural Policies in the Low-Data Regime

## Abstract

The progress in reinforcement learning algorithm development is at one of its highest points starting from the initial study that enabled sequential decision making from high-dimensional observations. Currently, deep reinforcement learning research has had quite recent breakthroughs from learning without the presence of rewards to learning functioning policies without even knowing the rules of the game. In our paper we focus on the trends currently used in deep reinforcement learning algorithm development in the low-data regime. We theoretically show that the performance profiles of the algorithms developed for the high-data regime do not transfer to the low-data regime in the same order. We conduct extensive experiments in the Arcade Learning Environment and our results demonstrate that the baseline algorithms perform significantly better in the low-data regime compared to the set of algorithms that were initially designed and compared in the large-data region.

## 1 Introduction

Reinforcement learning research achieved high acceleration upon the proposal of the initial study on approximating the state-action value function via deep neural networks (Mnih et al., 2015). Following this initial study several different highly successful deep reinforcement learning algorithms have been proposed (Hasselt et al., 2016b; Wang et al., 2016; Hessel et al., 2018; 2021) from targeting different architectural ideas to employing estimators targeting overestimation, all of which were designed and tested in the high-data regime (i.e. two hundred million frame training).

An alternative recent line of research with an extensive of amount of publications focused on pushing the performance bounds of deep reinforcement learning policies in the low-data regime (Yarats et al., 2021; Ye et al., 2021; Kaiser et al., 2020; van Hasselt et al., 2019; Kielak, 2019) (i.e. with one hundred thousand environment interaction training). Several different unique ideas in current reinforcement learning research, from model-based reinforcement learning to increasing sample efficiency with observation regularization, gained acceleration in several research directions based on policy performance comparisons demonstrated in the Arcade Learning Environment 100K benchmark. However, we demonstrate that there is a significant overlooked assumption being made in this line of research without being explicitly discussed. This implicit assumption, that is commonly shared amongst a large collection of low-data regime studies carries a significant importance due to the fact that these studies shape future research directions with incorrect reasoning; hence, influencing the overall research efforts put in for particular research ideas for several years following. Thus, in our paper we target this implicit assumption and aim to answer the following questions:

- *How can we theoretically explain the relationship between asymptotic sample complexity versus the low-data regime sample complexity in deep reinforcement learning?*

- *How would the performance profiles of deep reinforcement learning algorithms designed for the high-data regime transform to the low-data regime?*

- *Can we expect the performance rank of algorithms to hold with variations on the number of samples used in policy training?*

Hence, to be able to answer the questions raised above in our paper we focus on sample complexity in deep reinforcement learning and make the following contributions:

- We provide theoretical foundation on the non-transferability of the performance profiles of deep reinforcement learning algorithms designed for the high-data regime to the low-data regime.

- We theoretically demonstrate that the performance profile has a non-monotonic relationship with the asymptotic sample complexity and the low-data sample complexity region. Furthermore, we prove that the sample complexity of distributional reinforcement learning is higher than the sample complexity of baseline deep $\mathcal{Q}$-network algorithms.

- We conduct large scale extensive experiments for a variety of deep reinforcement learning baseline algorithms in both the low-data regime and the high-data regime Arcade Learning Environment benchmark.

- We highlight that recent algorithms proposed and evaluated in the Arcade Learning Environment 100K benchmark are significantly affected by the implicit assumption on the relationship between performance profiles and sample complexity.

## 2 BACKGROUND AND PRELIMINARIES

### 2.1 DEEP REINFORCEMENT LEARNING

The reinforcement learning problem is formalized as a Markov Decision Process (MDP) represented as a tuple $\langle S, A, \mathcal{P}, \mathcal{R}, \gamma, \rho_0 \rangle$ where $S$ represents the state space, $A$ represents the set of actions, $\mathcal{P}$ represents the transition probability distribution $\mathcal{P}$ on $S \times A \times S$, $\mathcal{R} : S \times A \to \mathbb{R}$ represents the reward function, and $\gamma \in (0, 1]$ represents the discount factor. The aim in reinforcement learning is to learn an optimal policy $\pi(s, a)$ that maps state observations to actions $\pi : S \to \Delta(A)$ that will maximize expected cumulative discounted rewards.

$$R = \mathbb{E}_{a_t \sim \pi(s_t, \cdot)} \sum_t \gamma^t \mathcal{R}(s_t, a_t, s_{t+1}), \tag{1}$$

This objective is achieved by constructing a state-action value function that learns for each state-action pair the expected cumulative discounted rewards that will be obtained if action $a \in A$ is executed in state $s \in S$.

$$\mathcal{Q}(s_t, a_t) = \mathcal{R}(s_t, a_t, s_{t+1}) + \gamma \sum_{s_t} \mathcal{P}(s_{t+1}|s_t, a_t)\mathcal{V}(s_{t+1}). \tag{2}$$

In settings where the state space and/or action space is large enough that the state-action value function $\mathcal{Q}(s, a)$ cannot be held in a tabular form, a function approximator is used. Thus, for deep reinforcement learning the $\mathcal{Q}$-function is approximated via deep neural networks.

$$\theta_{t+1} = \theta_t + \alpha(\mathcal{R}(s_t, a_t, s_{t+1}) + \gamma \mathcal{Q}(s_{t+1}, \arg\max_a \mathcal{Q}(s_{t+1}, a; \theta_t); \theta_t)$$
$$- \mathcal{Q}(s_t, a_t; \theta_t))\nabla_{\theta_t} \mathcal{Q}(s_t, a_t; \theta_t).$$

**Dueling Architecture:** At the end of convolutional layers for a given deep $\mathcal{Q}$-Network, the dueling architecture outputs two streams of fully connected layers for both estimating the state values $\mathcal{V}(s)$ and the advantage $\mathcal{A}(s, a)$ for each action in a given state $s$.

$$\mathcal{A}(s, a) = \mathcal{Q}(s, a) - \max_a \mathcal{Q}(s, a) \tag{3}$$

In particular, the last layer of the dueling architecture contains the forward mapping

$$\mathcal{Q}(s, a; \theta, \alpha, \beta) = \mathcal{V}(s; \theta, \beta) + \big(\mathcal{A}(s, a; \theta, \alpha) - \max_{a' \in A} \mathcal{A}(s, a'; \theta, \alpha)\big) \tag{4}$$

where $\theta$ represents the parameters of the convolutional layers and $\alpha$ and $\beta$ represent the parameters of the fully connected layers outputting the advantage and state value estimates respectively.

**Distributional Reinforcement Learning:** The baseline distributional reinforcement learning algorithm C51 was proposed by Bellemare et al. (2017). The projected Bellman update for the $i^{\text{th}}$ atom is computed as

$$(\Phi\mathcal{T}\mathcal{Z}_\theta(s_t, a_t))_i = \sum_{j}^{\mathcal{N}-1} \big[1 - \frac{|[\mathcal{T}z_j]_{v_{\min}}^{v_{\max}} - z_i|}{\Delta z}\big]_0^1 \tau_j(s_{t+1}, \max_{a \in A} \mathbb{E}\mathcal{Z}_\theta(s_{t+1}, a)) \tag{5}$$

where $z_i = v_{\min} + i\Delta z : 0 \leq i < \mathcal{N}$ represents the set of atoms in categorical learning, and the atom probabilities are learnt as a parametric model

$$\tau_i(s_t, \max_{a \in A} \mathbb{E}\mathcal{Z}_\theta(s_t, a)) = \frac{e^{\theta_i(s_t, a_t)}}{\sum_j e^{\theta_j(s_t, a_t)}} \quad \text{where} \quad \Delta z := \frac{v_{\max} - v_{\min}}{\mathcal{N} - 1} \tag{6}$$

Following this baseline algorithm Dabney et al. (2018b) proposed the $\mathcal{QRDQN}$ algorithm to learn the quantile projection of the state-action value distribution

$$\mathcal{TZ}(s_t, a_t) = \mathcal{R}(s_t, a_t, s_{t+1}) + \gamma \mathcal{Z}(s_{t+1}, \arg\max_{a \in A} \mathbb{E}_{z \sim \mathcal{Z}(s_{t+1}, a_{t+1})}[z]) \tag{7}$$

with $s_{t+1} \sim \mathcal{P}(\cdot|s_t, a_t)$ where $\mathcal{Z} \in Z$ represents the quantile distribution of an arbitrary value function. Following this study Dabney et al. (2018a) proposed the IQN algorithm (i.e. implicit quantile networks) to learn the full quantile function instead of learning a discrete set of quantiles as in the $\mathcal{QRDQN}$ algorithm. The IQN algorithm objective is to minimize the loss function

$$\mathcal{L} = \frac{1}{\mathcal{K}} \sum_{i=1}^{\mathcal{K}} \sum_{j=1}^{\mathcal{K}'} \rho_\delta(\mathcal{R}(s_t, a_t, s_{t+1}) + \gamma \mathcal{Z}_{\delta_{j'}}(s_{t+1}, \arg\max_{a \in A} \mathcal{Q}_\beta(s_t, a_t)) - \mathcal{Z}_{\delta_i}(s_t, a_t)) \tag{8}$$

where $\rho_\delta$ represents the Huber quantile regression loss, and $\mathcal{Q}_\beta = \int_0^1 \mathcal{F}_{\mathcal{Z}}^{-1}(\delta)d\beta(\delta)$. Note that $\mathcal{Z}_\delta = \mathcal{F}_{\mathcal{Z}}^{-1}(\delta)$ is the quantile function of the random variable $\mathcal{Z}$ at $\delta \in [0, 1]$.

## 3 LOW-DATA REGIME VERSUS ASYMPTOTIC PERFORMANCE

The high-level message of our empirical results is that comparing the asymptotic performance of two reinforcement learning algorithms does not necessarily give useful information on their relative performance in the low-data regime. In this section we provide mathematical motivation for this claim in the setting of finite-horizon MDPs with linear function approximation. In particular, a finite horizon MDP is represented as a tuple $\langle S, A, \mathcal{P}, \mathcal{R}, \mathcal{H} \rangle$ where $S$ is the set of states, and $A$ represents the set of actions. For each timestep $t \in [\mathcal{H}] = \{1, \ldots, \mathcal{H}\}$, state $s$, and action $a$ the transition probability kernel $\mathcal{P}_t(s_{t+1}|s_t, a_t)$ gives the probability distribution over the next state, and the reward $\mathcal{R}_t(s_t, a_t, s_{t+1})$ gives the immediate rewards. A non-stationary policy $\pi = (\pi_1, \ldots, \pi_\mathcal{H})$ induces a state-action value function given by

$$\mathcal{Q}_t^\pi(s_t, a_t) = \mathcal{R}_t(s_t, a_t, s_{t+1}) + \mathbb{E}_{s_t \sim \mathcal{P}_t(s_{t+1}|s_t, a_t), a_t \sim \pi} \left[ \sum_{h=t+1}^{\mathcal{H}} \mathcal{R}_t(s_h, \pi_h(s_h), s_{h+1}) \Big| s_t, a_t \right] \tag{9}$$

where we let $\pi(s)$ be the action taken by the policy $\pi$ in state $s$, and the corresponding value function $\mathcal{V}_t^\pi(s_t) = \mathcal{Q}_t(s_t, \pi(s_t))$. The optimal non-stationary policy $\pi^*$ has value function $\mathcal{V}_t^*(s_t) = \mathcal{V}_t^{\pi^*}(s_t)$ satisfying

$$\mathcal{V}_t^*(s_t) = \sup_\pi \mathcal{V}_t^\pi(s_t). \tag{10}$$

The objective is to learn a sequence of non-stationary policies $\pi^k$ for $k \in \{1, \ldots, \mathcal{K}\}$ while interacting with an unknown MDP in order to minimize the regret, which is measured asymptotically over $\mathcal{K}$ episodes of length $\mathcal{H}$

$$\text{REGRET}(\mathcal{K}) = \sum_{k=1}^{\mathcal{K}} \left( \mathcal{V}_1^*(s_1^k) - \mathcal{V}_1^{\pi^k}(s_1^k) \right) \tag{11}$$

where $s_1^k \in S$ is the starting state of the $k$-th episode. In words, regret sums up the gap between the expected rewards obtained by the sequence of learned policies $\pi^k$ and those obtained by $\pi^*$ when learning for $\mathcal{K}$ episodes. In the linear function approximation setting there is a feature map $\phi_t : S \times A \to \mathbb{R}^{d_t}$ for each $t \in [\mathcal{H}]$ that sends a state-action pair $(s, a)$ to the $d_t$-dimensional vector $\phi_t(s, a)$. Then, the state-action value function $\mathcal{Q}_t(s_t, a_t)$ is parameterized by a vector $\theta_t \in d_t$ so that $\mathcal{Q}_t(\theta_t)(s_t, a_t) = \phi_t(s, a)^\top \theta_t$.

Recent theoretical work in this setting Zanette et al. (2020) gives an algorithm along with a lower bound that matches the regret achieved by the algorithm up to logarithmic factors.

**Theorem 3.1** (Zanette et al. (2020)). *Under appropriate normalization assumptions there is an algorithm that learns a sequence of policies $\pi^k$ achieving regret*

$$\text{REGRET}(\mathcal{K}) = \tilde{O}\left(\sum_{t=1}^{\mathcal{H}} d_t \sqrt{\mathcal{K}} + \sum_{t=1}^{\mathcal{H}} \sqrt{d_t}\mathcal{I}\mathcal{K}\right), \tag{12}$$

*where $\mathcal{I}$ is the intrinsic Bellman error. Furthermore, this regret bound is optimal for this setting up to logarithmic factors in $d_t, \mathcal{K}$ and $\mathcal{H}$ whenever $\mathcal{K} = \Omega((\sum_{t=1}^{\mathcal{H}} d_t)^2)$, in the sense that for any level of intrinsic Bellman error $\mathcal{I}$, there exists a class of MDPs where any algorithm achieves at least as much regret on at least one MDP in the class.*

Utilizing this theorem we can then prove the following proposition on the relationship between the performance in the asymptotic and low-data regimes.

**Proposition 3.2.** *For every $1 > \alpha > \beta > 0$, there exist two thresholds $\mathcal{U} > \mathcal{L} > 1$, and a class of finite-horizon MDPs and feature maps $\phi_t$ each of dimension $d_t$ such that*

1. *Every algorithm receives regret at least $\text{REGRET}(\mathcal{K}) = \tilde{\Omega}\left(\alpha\mathcal{K}\right)$ after $\mathcal{K} \leq \mathcal{L}$ episodes.*

2. *There exists an algorithm receiving regret $\text{REGRET}(\mathcal{K}) = \tilde{O}\left(\beta\mathcal{K}\right)$ after $\mathcal{K} \geq \mathcal{U}$ episodes.*

*Proof.* Let $\mathcal{I} = \frac{\beta}{\sum_{t=1}^{\mathcal{H}} \sqrt{d_t}}$ and apply the lower bound of Theorem 3.1 with intrinsic Bellman error $\mathcal{I}$. Let $\mathcal{L} = O(\frac{1}{(\alpha-\beta)^2}(\sum_{t=1}^{\mathcal{H}} d_t)^2)$. Then after $\mathcal{K}$ episodes for $\mathcal{K} \leq \mathcal{L}$, the regret of any algorithm on the class of MDPs guaranteed by the theorem is at least

$$\text{REGRET}(\mathcal{K}) = \tilde{\Omega}\left(\sum_{t=1}^{\mathcal{H}} d_t \sqrt{\mathcal{K}} + \sum_{t=1}^{\mathcal{H}} \sqrt{d_t}\mathcal{I}K\right) = \tilde{\Omega}\left(\sum_{t=1}^{\mathcal{H}} d_t \sqrt{\mathcal{K}} + \beta\mathcal{K}\right)$$

$$= \tilde{\Omega}\left((\alpha - \beta)\sqrt{\mathcal{K}}\left(\frac{1}{\alpha - \beta}\sum_{t=1}^{\mathcal{H}} d_t\right) + \beta\mathcal{K}\right) \geq \tilde{\Omega}\left((\alpha - \beta)\mathcal{K} + \beta\mathcal{K}\right)$$

$$= \tilde{\Omega}\left(\alpha\mathcal{K}\right)$$

where the second equality follows from the choice of $\mathcal{I}$, and the inequality from the fact that $\mathcal{K} \leq \mathcal{L} = O(\frac{1}{(\alpha-\beta)^2}(\sum_{t=1}^{\mathcal{H}} d_t)^2)$. Next fix any constant $\epsilon > 0$ and let $\mathcal{U} = \Omega((\sum_{t=1}^{\mathcal{H}} d_t)^{\frac{2}{1-2\epsilon}})$. Then for $\mathcal{K} \geq \mathcal{U}$ episodes the algorithm guaranteed by Theorem 3.1 achieves regret

$$\text{REGRET}(\mathcal{K}) = \tilde{O}\left(\sum_{t=1}^{\mathcal{H}} d_t \sqrt{\mathcal{K}} + \sum_{t=1}^{\mathcal{H}} \sqrt{d_t}\mathcal{I}\mathcal{K}\right) = \tilde{O}\left(\sum_{t=1}^{\mathcal{H}} d_t \sqrt{\mathcal{K}} + \beta\mathcal{K}\right)$$

$$\leq \tilde{O}\left(\mathcal{K}^{1-\epsilon} + \beta\mathcal{K}\right) = \tilde{O}\left(\beta\mathcal{K}\right)$$

where the inequality follows from the bound $\mathcal{K} \geq \mathcal{U} = \Omega((\sum_{t=1}^{\mathcal{H}} d_t)^{\frac{2}{1-2\epsilon}})$. $\qquad\square$

Intuitively Proposition 3.2 shows that in the linear function approximation setting, the gap between performance in the low-data regime ($\mathcal{K} \leq \mathcal{L}$ episodes) and the high-data/asymptotic regime ($\mathcal{K} \geq \mathcal{U}$ episodes) can be arbitrarily large. Thus, comparisons between algorithms in the asymptotic/high-data regime are not informative when trying to understand algorithm performance with limited data.

## 4 MEAN ESTIMATION VERSUS LEARNING THE DISTRIBUTION

To obtain theoretical insight into the larger sample complexity exhibited by distributional reinforcement learning we consider the fundamental comparison between learning the distribution of a random variable $\mathcal{X}$ versus only learning the mean $\mathbb{E}[\mathcal{X}]$. In base distributional reinforcement learning the goal is to learn a distribution over state-action values that has finite support. Thus, to get a fundamental understanding of the additional cost of distributional reinforcement learning, we compare the sample complexity of learning the distribution of a finitely supported random variable with that of estimating the mean.

**Proposition 4.1.** *Let $\mathcal{X}$ be a real-valued random variable with support on exactly $k$ known values. Further, assume $|\mathcal{X}| < 1$ and let $\epsilon > 0$. Any algorithm that learns the distribution $\mathbb{P}(\mathcal{X})$ within total variation distance $\epsilon$ requires $\Omega(k/\epsilon^2)$ samples, while there exists an algorithm to estimate $\mathbb{E}[\mathcal{X}]$ to within error $\epsilon$ using only $O(1/\epsilon^2)$ samples.*

*Proof.* See appendix for the full proof. $\square$

Although Proposition 4.1 proves that distributional reinforcement learning has an intrinsically higher sample complexity than that of standard $\mathcal{Q}$-learning, it does not provide insights into the comparison of an error of $\epsilon$ in the mean with an error of $\epsilon$ in total variation distance. Hence, the following proposition demonstrates a precise justification of the comparison: whenever there are two different actions where the true mean state-action values are within $\epsilon$, an approximation error of $\epsilon$ in total variation distance for the state-action value distribution of one of the actions is sufficient to reverse the order of the means.

**Proposition 4.2.** *Fix a state $s$ and consider two actions $a, a'$. Let $\mathcal{X}(s, a)$ be the random variable distributed as the true state-action value distribution of $(s, a)$, and $\mathcal{X}(s, a')$ be the same for $(s, a')$. Suppose that $\mathbb{E}[\mathcal{X}(s, a)] = \mathbb{E}[\mathcal{X}(s, a')] + \epsilon$. Then there is a random variable $\mathcal{Y}$ such that*

$$d_{TV}(\mathcal{Y}, \mathcal{X}(s, a)) \leq \epsilon \text{ and } \mathbb{E}[\mathcal{X}(s, a')] \geq \mathbb{E}[\mathcal{Y}].$$

*Proof.* Let $\tau^* \in \mathbb{R}$ be the infimum

$$\tau^* = \inf\{\tau \in \mathbb{R} \mid \mathbb{P}[\mathcal{X}(s, a) \geq \tau] = \epsilon\} \tag{13}$$

i.e. $\tau^*$ is the first point in $\mathbb{R}$ such that $\mathcal{X}(s, a)$ takes values at least $\tau^*$ with probability exactly $\epsilon$. Next let the random variable $\mathcal{Y}$ be defined by the following process. First, sample the random variable $\mathcal{X}(s, a)$. If $\mathcal{X}(s, a) \geq \tau^*$, then output $\tau^* - 1$. Otherwise, output the sampled value of $\mathcal{X}(s, a)$.

Observe that the probability distributions of $\mathcal{Y}$ and $\mathcal{X}(s, a)$ are identical except at the point $\tau^* - 1$ and on the interval $[\tau^*, \infty)$. Let $\mu$ be the Lebesgue measure on $\mathbb{R}$. By construction of $\mathcal{Y}$ the total variation distance is given by

$$\begin{aligned}
d_{TV}(\mathcal{Y}, \mathcal{X}) &= \frac{1}{2} \int_{\mathbb{R}} \left| \mathbb{P}[\mathcal{X}(s, a) = z] - \mathbb{P}[\mathcal{Y} = z] \right| d\mu(z) \\
&= \frac{1}{2} \left| \mathbb{P}[\mathcal{X}(s, a) = \tau^* - 1] - \mathbb{P}[\mathcal{Y} = \tau^* - 1] \right| \\
&\quad + \frac{1}{2} \int_{[\tau^*, \infty)} \left| \mathbb{P}[\mathcal{X}(s, a) = z] - \mathbb{P}[\mathcal{Y} = z] \right| d\mu(z) = \frac{\epsilon}{2} + \frac{\epsilon}{2} = \epsilon.
\end{aligned}$$

Next note that the expectation of $\mathcal{Y}$ is given by

$$\begin{aligned}
\mathbb{E}[\mathcal{Y}] &= \epsilon(\tau^* - 1) + \int_{(-\infty, \tau^*]} z \mathbb{P}[\mathcal{X}(s, a) = z] d\mu(z) \\
&= \epsilon(\tau^* - 1) + \int_{\mathbb{R}} z \mathbb{P}[\mathcal{X}(s, a) = z] d\mu(z) - \int_{(\tau^*, \infty]} z \mathbb{P}[\mathcal{X}(s, a) = z] d\mu(z) \\
&\leq \epsilon(\tau^* - 1) + \mathbb{E}[\mathcal{X}(s, a)] - \epsilon\tau^* \\
&= \mathbb{E}[\mathcal{X}(s, a)] - \epsilon
\end{aligned}$$

where the inequality follows from the fact that $\mathcal{X}$ takes values larger than $\tau^*$ with probability $\epsilon$. $\square$

To summarize, Proposition 4.2 shows that, in the case where the mean state-action values are within $\epsilon$, unless the state-action value distribution is learned to within total-variation distance $\epsilon$, the incorrect action may be selected by the distributional reinforcement learning policy. Therefore, it is natural to compare the sample complexity of learning the state-action value distribution to within total-variation distance $\epsilon$ with the sample complexity of simply learning the mean to within error $\epsilon$, as is done in Proposition 4.1.

## 5 SAMPLE COMPLEXITY WITH UNKNOWN SUPPORT

The setting considered in Proposition 4.1 most readily applies to the base distributional reinforcement learning algorithm C51, which attempts to directly learn a discrete distribution with known support in order to approximate the state-action value distribution. However, further advances in distributional reinforcement learning including $\mathcal{Q}$RD$\mathcal{Q}$N and I$\mathcal{Q}$N do away with the assumption that the support of the distribution is known. This allows a more flexible representation in order to more accurately represent the true distribution on state-action values, but, as we will show, potentially leads to a further increase in the sample complexity. The $\mathcal{Q}$RD$\mathcal{Q}$N algorithm models the distribution of state-action values as a uniform mixture of $\mathcal{N}$ Dirac deltas on the reals i.e. $\mathcal{Z}(s,a) = \frac{1}{\mathcal{N}} \sum_{i=1}^{\mathcal{N}} \delta_{\theta_i(s,a)}$, where $\theta_i(s,a) \in \mathbb{R}$ is a parametric model.

**Proposition 5.1.** *Let $\mathcal{N} > \mathcal{M} \geq 2$, $\epsilon > \frac{\mathcal{M}}{4\mathcal{N}}$, and $\theta_i \in \mathbb{R}$ for $i \in [\mathcal{N}]$. The number of samples required to learn a distribution of the form $\mathcal{Z} = \frac{1}{\mathcal{N}} \sum_{i=1}^{\mathcal{N}} \delta_{\theta_i}$ to within total variation distance $\epsilon$ is $\Omega\left(\frac{\mathcal{M}}{\epsilon^2}\right)$.*

*Proof.* Let $\mathcal{M} \geq 2$ and $\mathcal{D} = \{1, 2, \cdots, \mathcal{M}\} \subseteq \mathbb{R}$. First we will show that any distribution $p(z)$ supported on $z \in \mathcal{D}$ is within total-variation distance $\frac{k}{4\mathcal{N}}$ of a distribution of a random variable of the form $\mathcal{Z} = \frac{1}{\mathcal{N}} \sum_{i=1}^{\mathcal{N}} \delta_{\theta_i}$ for numbers $\theta_i \in \mathcal{D}$. Indeed we can construct such a distribution as follows. First let $\tilde{p}(z)$ be the rounded distribution obtained by rounding each probability $p(z)$ to the nearest integer multiple of $\frac{1}{\mathcal{N}}$. The total variation distance between $p(z)$ and $\tilde{p}(z)$ is given by

$$\frac{1}{2} \sum_{z=1}^{\mathcal{M}} |p(z) - \tilde{p}(z)| \leq \frac{1}{2} \sum_{z=1}^{\mathcal{M}} \frac{1}{2\mathcal{N}} \leq \frac{\mathcal{M}}{4\mathcal{N}}. \tag{14}$$

Next partition the set of $\theta_i$ into $\mathcal{M}$ groups $\mathcal{G}_1, \mathcal{G}_2, \ldots, \mathcal{G}_{\mathcal{M}}$, where group $\mathcal{G}_z$ has size $\mathcal{N}\tilde{p}(z)$ (this size is an integer by construction of $\tilde{p}$). Finally, for each $\theta_i \in \mathcal{G}_z$ assign $\theta_i = z$. Thus for $\mathcal{Z} = \frac{1}{\mathcal{N}} \sum_{i=1}^{\mathcal{N}} \delta_{\theta_i}$ we have for each $z \in \mathcal{D}$

$$\mathbb{P}[\mathcal{Z} = z] = \frac{1}{\mathcal{N}} \sum_{i=1}^{\mathcal{N}} \mathbb{1}[\theta_i = z] = \frac{1}{\mathcal{N}} |\mathcal{G}_z| = \tilde{p}(z). \tag{15}$$

Therefore, any distribution $p(z)$ can be approximated to within total variation distance $\frac{\mathcal{M}}{4\mathcal{N}}$ by a distribution $\mathcal{Z}$ of the prescribed form. Thus, by the sample complexity lower bounds for learning a discrete distribution with known support, for any $\epsilon > \frac{\mathcal{M}}{4\mathcal{N}}$ at least $\frac{\mathcal{M}}{\epsilon^2}$ samples are required to learn a distribution of the form $\mathcal{Z} = \frac{1}{\mathcal{N}} \sum_{i=1}^{\mathcal{N}} \delta_{\theta_i}$. $\qquad\square$

Depending on the choice of parameters, the lower bound in Proposition 5.1 can be significantly larger than that of Proposition 4.1. For example if the desired approximation error is $\epsilon = \frac{1}{8}$ one can take $\mathcal{M} = \frac{\mathcal{N}}{2}$. In this case if the value of $k$ in Proposition 4.1 satisfies $k = o(\mathcal{N})$, then the sample complexity in Proposition 5.1 is asymptotically larger than that of Proposition 4.1.

## 6 LARGE SCALE EXPERIMENTAL INVESTIGATION

The experiments are conducted in the Arcade Learning Environment (ALE) (Bellemare et al., 2013). The Double $\mathcal{Q}$-learning algorithm is trained via Double Deep $\mathcal{Q}$-Network (Hasselt et al., 2016a) initially proposed by van Hasselt (2010). The dueling algorithm is trained via Wang et al. (2016). The prior algorithm refers to the prioritized experience replay algorithm proposed by Schaul et al. (2016). The distributional reinforcement learning policies are trained via the C51 algorithm, I$\mathcal{Q}$N and $\mathcal{Q}$RD$\mathcal{Q}$N. To provide a complete picture of the sample complexity we conducted our experiments in both low-data, i.e. the Arcade Learning Environment 100K benchmark, and high data regime, i.e. baseline 200 million frame training. All of the results are reported with the standard error of the mean in all of the tables and figures in the paper. The experiments are run with JAX (Bradbury et al., 2018), with Haiku as the neural network library, Optax (Hessel et al., 2020) as the optimization library, and RLax for the reinforcement learning library (Babuschkin et al., 2020).

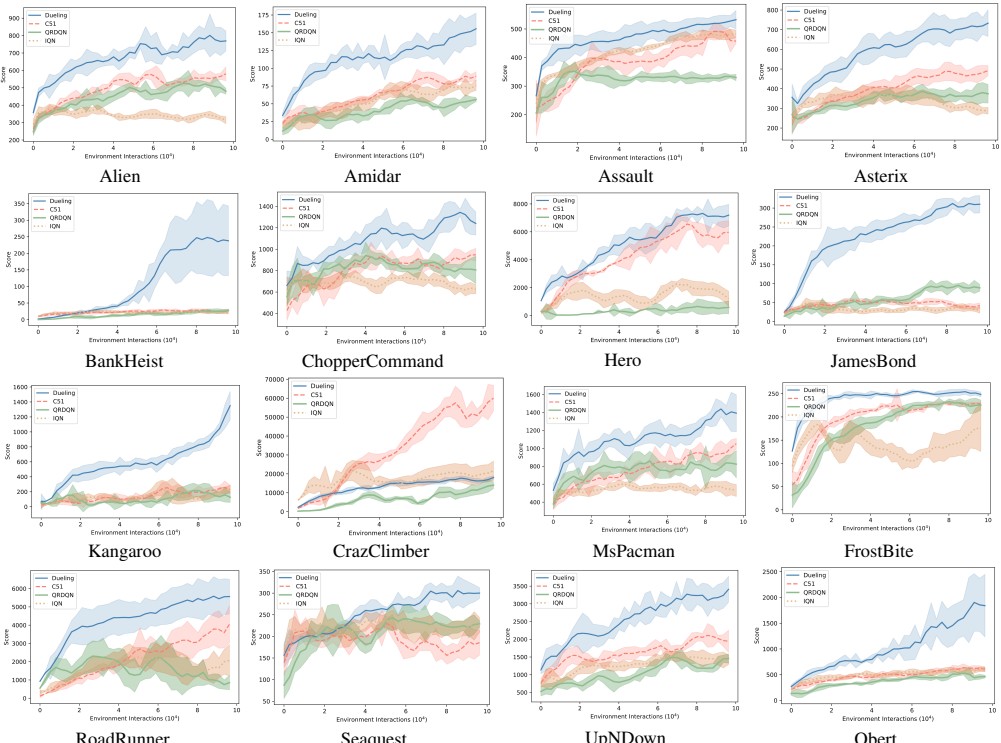

Figure 1: The learning curves of Alien, Amidar, Asterix, BankHeist, ChopperCommand, Hero, CrazyClimber, JamesBond, Kangaroo, MsPacman, FrostBite, Qbert, RoadRunner, Seaquest and UpNDown with dueling architecture, C51, IQN and QRDQN algorithms in the Arcade Learning Environment with 100K environment interaction training. See appendix for the full learning curves.

More details on the hyperparameters and direct references to the implementations can be found in the appendix. Note that human normalized score is computed as follows:

$$\text{Score}_{\text{HN}} = \frac{\text{Score}_{agent} - \text{Score}_{random}}{\text{Score}_{human} - \text{Score}_{random}} \tag{16}$$

Figure 1 reports learning curves for the IQN, QRDQN, dueling architecture and C51 for every MDP in the Arcade Learning Environment low-data regime 100K benchmark. These results demonstrate that the simple base algorithm dueling performs significantly better than any distributional algorithm when the training samples are limited. For a fair, direct and transparent comparison we kept the hyperparameters for the baseline algorithms in the low-data regime exactly the same with the DRQ$^{\text{ICLR}}$ paper (see appendix for the full list and high-data regime hyperparameter settings). Note that the DRQ algorithm uses the dueling architecture without any distributional reinforcement learning. One intriguing takeaway from the results provided in Table 1 and the Figure 4[1] is the fact that the simple base algorithm dueling performs 15% better than the DRQ$^{\text{NeurIPS}}$ implementation, and 11% less than the DRQ$^{\text{ICLR}}$ implementation. Note that the original paper of the DRQ$^{\text{ICLR}}$ algorithm provides comparison only to data-efficient Rainbow (DER) (van Hasselt et al., 2019) which inherently uses distributional reinforcement learning. The fact that the original paper that proposed data augmentation for deep reinforcement learning (i.e. DRQ$^{\text{ICLR}}$) on top of the dueling architecture did not provide comparisons with the pure simple base dueling architecture (Wang et al., 2016) resulted in inflated performance profiles for the DRQ$^{\text{ICLR}}$ algorithm.

More intriguingly, the comparisons provided in the DRQ$^{\text{ICLR}}$ paper to the DER and OTR algorithms report the performance gained by DRQ$^{\text{ICLR}}$ over DER is 82% and over OTR is 35%. However, if a

---

[1]DER$^{2021}$ refers to the re-implementation with random seed variations of the original paper data-efficient Rainbow (i.e. DER$^{2019}$) by van Hasselt et al. (2019). OTR refers to further implementation of the Rainbow algorithm by Kielak (2019). DRQ$^{\text{NeurIPS}}$ refers to the re-implementation of the original DRQ algorithm published in ICLR as a spotlight presentation with the goal of achieving reproducibility with variation on the number of random seeds (Agarwal et al., 2021).

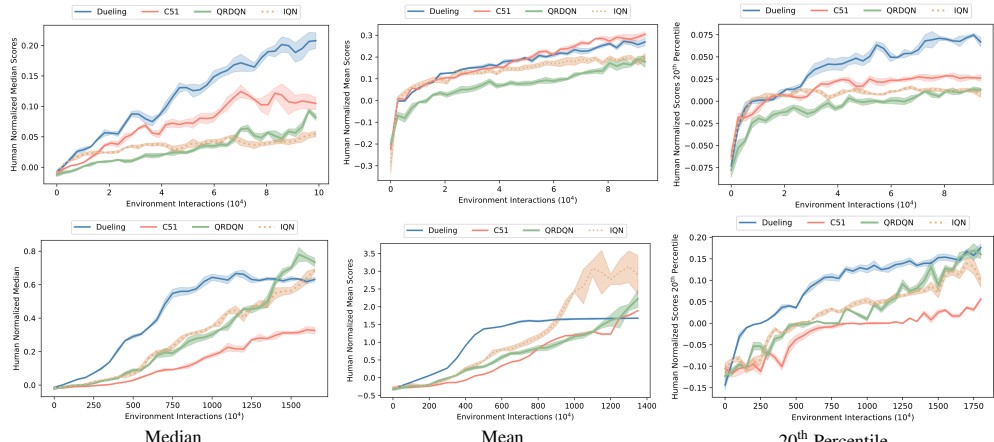

Figure 2: Up: Human normalized median, mean and 20[th] percentile results for the dueling algorithm, C51, I$\mathcal{Q}$N and $\mathcal{Q}$RD$\mathcal{Q}$N in the Arcade Learning Environment 100K benchmark. Down: Human normalized median, mean, and 20[th] percentile results for the dueling algorithm, C51, I$\mathcal{Q}$N and $\mathcal{Q}$RD$\mathcal{Q}$N in the high-data regime towards 200 million frame.

Table 1: Large scale comparison of $\mathcal{Q}$-based deep reinforcement learning algorithms with human normalized mean, median and 20[th] percentile results in the Arcade Learning Environment 100K benchmark for D$\mathcal{Q}$N (Mnih et al., 2015), deep Double-$\mathcal{Q}$ learning (Hasselt et al., 2016a), dueling architecture (Wang et al., 2016), Prior (Schaul et al., 2016) C51, $\mathcal{Q}$RD$\mathcal{Q}$N and I$\mathcal{Q}$N.

| Algorithms | Human Normalized Median | Human Normalized Mean | 20[th] Percentile |
|---|---|---|---|
| D$\mathcal{Q}$N | 0.0481±0.0036 | 0.1535±0.0119 | 0.0031±0.0032 |
| Double-$\mathcal{Q}$ | 0.0920±0.0181 | **0.3169±0.0196** | 0.0341±0.0042 |
| Dueling | **0.2304±0.0061** | 0.2923±0.0060 | **0.0764±0.0037** |
| C51 | 0.0941±0.0081 | 0.3106±0.0199 | 0.0274±0.0024 |
| $\mathcal{Q}$RD$\mathcal{Q}$N | 0.0820±0.0037 | 0.2171±0.0098 | 0.0189±0.0031 |
| I$\mathcal{Q}$N | 0.0528±0.0058 | 0.2050±0.0123 | 0.0091±0.0011 |
| Prior | 0.0840±0.0018 | 0.2792±0.0123 | 0.0267±0.0042 |

direct comparison is made to the simple dueling algorithm as Table 1 demonstrates with the exact hyperparameters used as in the DR$\mathcal{Q}$[ICLR] paper the performance gain is utterly restricted to 11%. Moreover, when it is compared to the reproduced results of DR$\mathcal{Q}$[NeurIPS] it turns out that there is a performance decrease due to utilizing the DR$\mathcal{Q}$ algorithm over dueling architecture. Thus, the fact that our paper provides foundations on the non-transferability of the performance profile results from large-data regime to low-data regime can influence future research to have more concrete and accurate performance profiles for algorithm development in the low-data regime.

Table 1 reports the human normalized median, human normalized mean, and human normalized 20[th] percentile results over all of the MDPs from the 100K Arcade Learning Environment benchmark for D$\mathcal{Q}$N, Double-$\mathcal{Q}$, dueling, C51, $\mathcal{Q}$RD$\mathcal{Q}$N, I$\mathcal{Q}$N and prior. One important takeaway from the results reported in the Table 1 is the fact that one particular algorithm performance profile in 200 million frame training will not directly transfer to the low-data region. Figure 2 reports the learning curves of human normalized median, human normalized mean and human normalized 20[th] percentile for the dueling algorithm, C51, $\mathcal{Q}$RD$\mathcal{Q}$N, and I$\mathcal{Q}$N in the low-data region. These results once more demonstrate that the performance profile of the simple base algorithm dueling is significantly better than any other distributional reinforcement learning algorithm when the number of environment interactions are limited.

The left and center plots of Figure 3 report regret curves corresponding to the theoretical analysis in Proposition 3.2 for various choices of the feature dimensionality $d$ and the intrinsic Bellman error $\mathcal{I}$. In particular, the left plot shows the low-data regime where the number of episodes $\mathcal{K} < 1000$, while the right plot shows the high-data regime where $\mathcal{K}$ is as large as 500000. Notably, the relative ordering of the regret across the different choices of $d$ and $\mathcal{I}$ is completely reversed in the high-data

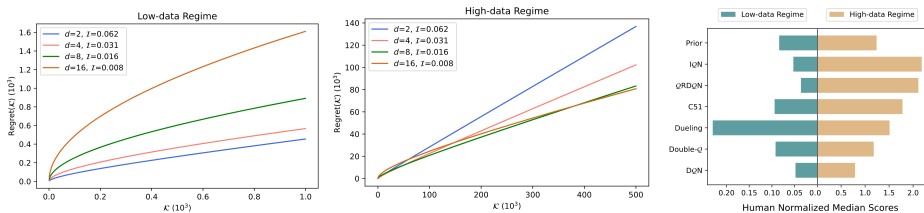

Figure 3: Left: Regret in the low-data regime. Center: Regret in the high-data regime. Right: Distributional vs baseline $\mathcal{Q}$ comparison of algorithms that were proposed and developed in the high-data regime in the Arcade Learning Environment in both high-data regime and low-data regime.

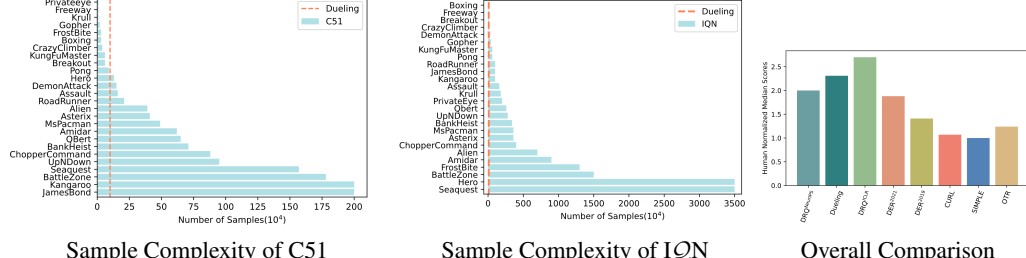

Figure 4: Left: Number of samples (i.e. environment interactions) required by the base distributional reinforcement learning algorithm to achieve the performance level achieved by the dueling algorithm. Center: Number of samples required by I$\mathcal{Q}$N to achieve the performance level achieved by dueling. Right: Overall comparison of algorithms recently developed in the low-data regime ALE 100K benchmark to the dueling algorithm that were designed in the high-data region.

regime when compared to the low-data regime. Figure 4 reports results on the number of samples required for training with the baseline distributional reinforcement learning algorithm to reach the same performance levels achieved by the dueling algorithm for each individual MDP from the Arcade Learning Environment low-data regime benchmark. These results once more demonstrate that to reach the same performance levels with the dueling algorithm, the baseline distributional reinforcement learning algorithm requires orders of magnitude more samples to train on. As discussed in Section 5, more complex representations for broader classes of distributions come at the cost of a higher sample complexity required for learning.

One intriguing fact is that the original SimPLE paper provides a comparison in the low-data regime of their proposed algorithm with the Rainbow algorithm which is essentially an algorithm that is designed in the high-data region by having the implicit assumption that the state-of-the art performance profile must transfer linearly to the low-data region. These instances of implicit assumptions also occur in DR$\mathcal{Q}^{\text{ICLR}}$, CURL, SPR and Efficient-Zero even when comparisons are made for more advanced algorithms such as MuZero.

## 7 CONCLUSION

In this paper we aimed to answer the following questions: (i) *Do the performance profiles of deep reinforcement learning algorithms designed for certain data regimes translate approximately linearly to a different sample complexity region?*, and (ii) *What is the underlying theoretical relationship between the performance profiles and sample complexity regimes?* To be able to answer these questions we provide theoretical investigation on the sample complexity of the baseline deep reinforcement learning algorithms. We conduct extensive experiments both in the low-data region 100K Arcade Learning Environment and high-data regime baseline 200 million frame training. Our results demonstrate that the performance profiles of deep reinforcement learning algorithms do not have a monotonic relationship across sample complexity regimes. The implicit assumption of the monotonic relationship of the performance characteristics and the sample complexity regions that exists in many recent state-of-the-art studies has been overly exploited. Thus, our results demonstrate that several baseline $\mathcal{Q}$ algorithms are almost as high performing as recent variant algorithms that have been proposed and shown as the state-of-the-art.

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

# A APPENDIX

## A.1 RESULTS ON THE COMPLETE LIST OF GAMES FROM ARCADE LEARNING ENVIRONMENT 100K BASELINE

Table 2 reports the average scores obtained by the human player, random player, baseline $\mathcal{Q}$-based algorithm dueling architecture, baseline distributional reinforcement learning algorithm C51, $\mathcal{Q}$RD$\mathcal{Q}$N and I$\mathcal{Q}$N across all the games in the Arcade Learning Environment 100K baseline. These results once more demonstrate that the baseline $\mathcal{Q}$-based algorithm performs significantly better than any distributional reinforcement learning algorithm as has also been explained in detail in Section 5.

Table 2: Average returns for human, random, dueling Wang et al. (2016), C51, $\mathcal{Q}$RD$\mathcal{Q}$N and I$\mathcal{Q}$N across all the games in the Arcade Learning Environment 100K benchmark.

| Games | Human | Random | C51 | $\mathcal{Q}$RD$\mathcal{Q}$N | I$\mathcal{Q}$N | Dueling |
|---|---|---|---|---|---|---|
| Alien | 7127.7 | 227.8 | 547.16 | 509.57 | 330.81 | **705.58** |
| Amidar | 1719.5 | 5.8 | 78.41 | 55.70 | 74.98 | **199.31** |
| Assault | 742.0 | 222.4 | 465.30 | 314.58 | 488.55 | **503.82** |
| Asterix | 8503.3 | 210.0 | 475.90 | 367.32 | 286.26 | **705.16** |
| BankHeist | 753.1 | 14.2 | 22.81 | 21.53 | 18.17 | **243.19** |
| BattleZone | 37187.5 | 2360.0 | 2728.52 | 6238.27 | 3105.70 | **6880.37** |
| Boxing | 12.1 | 0.1 | 9.60 | 2.03 | **12.41** | 1.68 |
| Breakout | 30.5 | 1.7 | 11.35 | **16.50** | 15.09 | 8.28 |
| ChopperCommand | 7387.8 | 811.0 | 831.83 | 752.51 | 629.04 | **1313.90** |
| CrazyClimber | 35829.4 | 10780.5 | **71776.14** | 21366.42 | 22649.44 | 17039.44 |
| DemonAttack | 1971.0 | 152.1 | 789.09 | 198.01 | **1035.17** | 694.42 |
| Freeway | 29.6 | 0.0 | **20.42** | 5.98 | 19.37 | 5.93 |
| FrostBite | 4334.7 | 65.2 | 215.25 | 218.11 | 192.33 | **259.18** |
| Gopher | 2412.5 | 257.6 | **791.83** | 576.19 | 466.81 | 429.85 |
| Hero | 30826.4 | 1027.0 | 7097.42 | 1108.44 | 1322.63 | **8210.53** |
| Jamesbond | 302.8 | 29.0 | 43.85 | 108.71 | 26.23 | **296.46** |
| Kangaroo | 3035.0 | 52.0 | 301.01 | 120.60 | 294.46 | **1914.86** |
| Krull | 2665.5 | 1598.0 | **3744.04** | 2040.50 | 2319.74 | 2867.78 |
| KungFuMaster | 22736.3 | 258.5 | 6877.62 | **11574.02** | 1526.76 | 5367.90 |
| Mspacman | 6951.6 | 307.3 | 917.78 | 749.29 | 533.98 | **1355.21** |
| Pong | 14.6 | -20.7 | **11.17** | -7.49 | -10.86 | -4.20 |
| PrivateEye | 69571.3 | 24.9 | -103.30 | -6.32 | 33.83 | **100.00** |
| Qbert | 13455.0 | 163.9 | 528.30 | 590.05 | 582.72 | **1710.23** |
| RoadRunner | 7845.0 | 11.5 | 3993.34 | 400.59 | 1202.20 | **6031.80** |
| Seaquest | 42054.7 | 68.4 | 163.69 | 183.25 | 213.87 | **351.10** |
| UpNdDown | 11693.2 | 533.4 | 1970.28 | 1622.67 | 1552.27 | **3553.12** |

Figure 5 reports the learning curves of the complete list of the games in the Arcade Learaning Environment 100K benchmark; in particular, for Alien, Amidar, Asterix, BankHeist, BattleZone, Boxing, Breakout, ChopperCommand, Hero, CrazyClimber, JamesBond, Kangaroo, PrivateEye, MsPacman, FrostBite, Qbert, RoadRunner, Seaquest, Pong, Gopher, DemonAttack, Krull, and UpNDown with dueling architecture Wang et al. (2016), C51, I$\mathcal{Q}$N and $\mathcal{Q}$RD$\mathcal{Q}$N algorithms with 100K environment interaction training. The learning curves reported in Figure 5 demonstrate that the number of samples required to obtain the performance level achieved via the simple base dueling architecture is significantly higher for any distributional reinforcement learning algorithm. Note that the distributional reinforcement learning algorithm C51 represents the state-action value distribution as a discrete probability distribution supported on 51 fixed atoms evenly spaced between a pre-specified minimum and maximum value. In contrast, QR-D$\mathcal{Q}$N represents the value distribution as the uniform distribution over a larger number of atoms with variable positions on the real line. Thus, QR-D$\mathcal{Q}$N is able to more accurately approximate a broader class of state-action value distributions. Finally, I$\mathcal{Q}$N parameterizes the quantile function of the state-action value distribution via a deep neural network, leading to a yet more flexible representation of the state-action value distribution. As discussed in Section 5, more complex representations for broader classes of distributions come at the cost of a higher sample complexity required for learning.

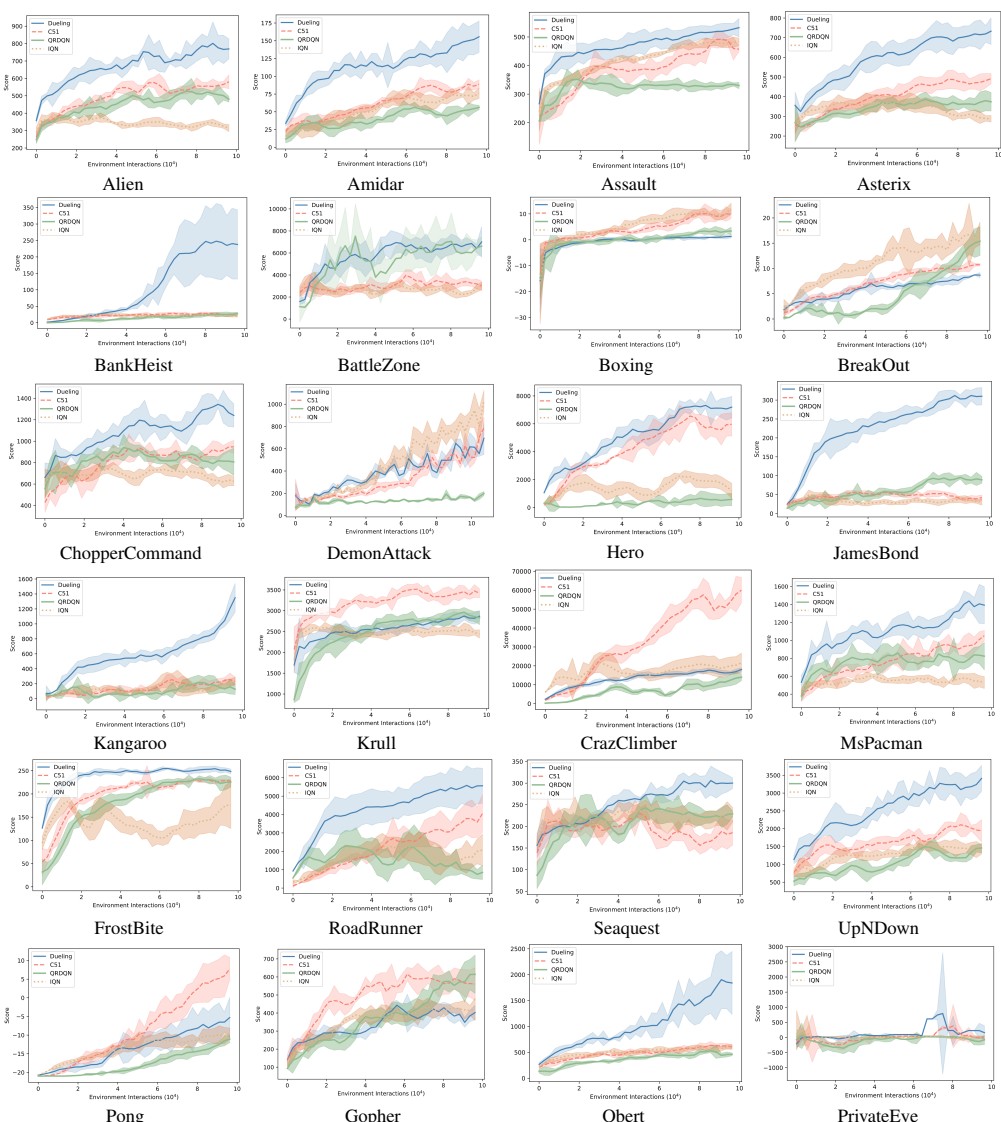

Figure 5: The learning curves of Alien, Amidar, Asterix, BankHeist, BattleZone, Boxing, Breakout, ChopperCommand, Hero, CrazyClimber, JamesBond, Kangaroo, PrivateEye, MsPacman, FrostBite, Qbert, RoadRunner, Seaquest, Pong, Gopher, DemonAttack, Krull, and UpNDown with dueling architecture Wang et al. (2016), C51, I$\mathcal{Q}$N and $\mathcal{Q}$RD$\mathcal{Q}$N algorithms in the Arcade Learning Environment with 100K environment interaction training.

## A.2 Reproducibility and Configuration Details

The hyperparameter settings of all of the algorithms in our paper, double-$\mathcal{Q}$, dueling, $\mathcal{Q}$RD$\mathcal{Q}$N, and I$\mathcal{Q}$N for the high-data region are exactly the same with the original papers that proposed these algorithms in the high-data region. See the hyperparameter settings in Hasselt et al. (2016) for double-$\mathcal{Q}$, Wang et al. (2016) for dueling architecture, Bellemare et al. (2017) for C51, Dabney et al. (2018a) for $\mathcal{Q}$RD$\mathcal{Q}$N, and Dabney et al. (2018b) for I$\mathcal{Q}$N.

For a fair and transparent comparison, we kept the hyperparameters exactly the same with the DR$\mathcal{Q}$[ICLR] paper for all of the baseline $\mathcal{Q}$ algorithms in the low-data region. Note that DR$\mathcal{Q}$ is an observation regularization study; hence the hyperparameters in the DR$\mathcal{Q}$ paper are specifically tuned for the purpose of the paper besides tuning for the Arcade Learning Environment 100K low-data regime. We did not tune any of the hyperparameters for the baseline algorithms (i.e. dueling

architecture and distributional reinforcement learning algorithms). Hence, it is even further possible to conduct hyperparameter tuning and get better performance profile results with the simple baseline dueling architecture. For the purpose of this paper we kept the hyperparameters exactly the same with the DR$\mathcal{Q}^{\text{ICLR}}$ paper. However, we would strongly encourage further research to conduct hyperparameter optimization to obtain better results from the baseline dueling architecture in the low-data regime.

Table 3: Hyperparameter settings and architectural details for the dueling algorithm, double-$\mathcal{Q}$ learning, C51, $\mathcal{Q}$RD$\mathcal{Q}$N, and I$\mathcal{Q}$N in the low-data regime of the Arcade Learning Environment.

| Hyperparameters | Settings |
|---|---|
| Grey-scaling | True |
| Observation down-sampling | (84, 84) |
| Action repetitions | 4 |
| Frames stacked | 4 |
| Batch Size | 32 |
| Update | Double-Q |
| Max Frames per episode | 108000 |
| Evaluation exploration epsilon | 0.01 |
| Min replay size for sampling | 1600 |
| Max gradient norm | 10 |
| Discount factor | 0.99 |
| Maximum absolute rewards | 1 |
| Training steps | 100000 |
| Evaluation steps | 125000 |
| Exploration epsilon decay frame fraction | 0.0125 |
| Gradient error bound | 0.03125 |
| Optimizer | Adam |
| Replay period every | 1 |
| n-step length | 10 |
| Exploration | $\epsilon$-greedy |
| $\epsilon$-decay | 5000 |
| Number of atoms | 51 |
| Number of quantiles | 201 |
| $v_{\max}$ | 10 |
| $\mathcal{Q}$-Network channels | 32,64,64 |
| $\mathcal{Q}$-Network filter size | $8 \times 8, 4 \times 4, 3 \times 3$ |
| $\mathcal{Q}$-Network stride | $(4, 4), (2, 2), (1, 1)$ |
| $\mathcal{Q}$-Network hidden units | 512 |

We have also tried the hyperparameter settings reported in the data efficient Rainbow (DER) paper for C51, I$\mathcal{Q}$N and $\mathcal{Q}$RD$\mathcal{Q}$N in the low-data regime. The performance results are provided in Table 4 for the hyperparameter settings of DER. As can be seen, the hyperparameter settings of DRQ$^{\text{ICLR}}$ gave better performance results also for C51, I$\mathcal{Q}$N and $\mathcal{Q}$RD$\mathcal{Q}$N in the low-data region. The results in Table 4 also align with the claims of the DER paper in which there was not been extensive hyperparameter tuning conducted to achieve the results provided, and it is possible to obtain better results by further hyperparameter tuning.

Table 4: Human normalized mean, human normalized median, and human normalized 20[th] percentile results for the C51 algorithm, $\mathcal{Q}$RD$\mathcal{Q}$N, and I$\mathcal{Q}$N in the low-data regime of the Arcade Learning Environment with the hyperparameter settings reported in the DER paper.

| Algorithms | Human Normalized Median | Human Normalized Mean | Human Normalized 20[th] Percentile |
|---|---|---|---|
| C51 | 0.0490±0.0038 | 0.1352±0.0057 | 0.0163±0.0029 |
| $\mathcal{Q}$RD$\mathcal{Q}$N | 0.0203±0.0033 | 0.0778±0.0101 | -0.0012±0.0053 |
| I$\mathcal{Q}$N | 0.0202±0.0020 | 0.0590±0.0139 | -0.0035±0.0031 |

### A.3 MEAN ESTIMATION VERSUS LEARNING THE DISTRIBUTION

**Proposition A.1** (Proposition 4.1). *Let $\mathcal{X}$ be a real-valued random variable with support on exactly $k$ known values. Further, assume $|\mathcal{X}| < 1$ and let $\epsilon > 0$. Any algorithm that learns the distribution $\mathbb{P}(\mathcal{X})$ within total variation distance $\epsilon$ requires $\Omega(k/\epsilon^2)$ samples, while there exists an algorithm to estimate $\mathbb{E}[\mathcal{X}]$ to within error $\epsilon$ using only $O(1/\epsilon^2)$ samples.*

*Proof.* Learning a distribution with known discrete support of size $k$ requires $\Omega(k/\epsilon^2)$ samples to achieve total variation distance at most $\epsilon$ with constant probability (Canonne, 2020). On the other hand, let $\mathcal{X}_1, \ldots, \mathcal{X}_n$ be independent samples of the random variable $\mathcal{X}$ and consider the sample mean

$$\bar{\mathcal{X}} = \frac{1}{n} \sum_{i=1}^{n} \mathcal{X}_i. \tag{17}$$

The expectation is given by $\mathbb{E}[\bar{\mathcal{X}}] = \mathbb{E}[\mathcal{X}]$ and the variance is $\sigma^2(\bar{\mathcal{X}}) = \frac{1}{n}\sigma^2(\mathcal{X})$. Further, since $|\mathcal{X}| < 1$ we have that $\sigma^2(X) < 1$ and so $\sigma^2(\bar{\mathcal{X}}) \leq \frac{1}{n}$. Hence, by Chebyshev's inequality

$$\mathbb{P}\left[|\bar{\mathcal{X}} - \mathbb{E}[\mathcal{X}]| > \epsilon\right] \leq \frac{1}{\epsilon^2 n}. \tag{18}$$

Thus with $n = O(\frac{1}{\epsilon^2})$ samples, $\bar{\mathcal{X}}$ is within $\epsilon$ of $\mathbb{E}[\mathcal{X}]$ with constant probability. $\square$

