# OpenReview forum: "A Large Scale Sample Complexity Analysis of Neural Policies in the Low-Data Regime"
_ICLR.cc/2023/Conference — Submitted to ICLR 2023_

### Official Review · Reviewer_88qX · 2022-10-24

**Confidence:** 4
**Correctness:** 3
**Technical Novelty And Significance:** 4
**Empirical Novelty And Significance:** 4
**Recommendation:** 8

**Clarity, Quality, Novelty And Reproducibility:**

The investigation is a very useful one, both from the theory and empirical sides. The paper is well-written and clear, with a nice and easy-to-follow flow. There may be other papers that have investigated similar questions, but to the best of my knowledge, no paper has investigated such questions in a focused manner and both from the theory and empirical sides, with empirical verifications focused on deep RL.

**Questions:**

- Are the theoretical results of Section 4 currently only proven in the linear FA case?

**Minor:**

- Sec 2.1, line 3 of Par 1: $\mathcal{R}$ represents the **expected** reward function. Later in your equations (e.g. Eq 1) you use $\mathcal{R}(s,a,s')$, thus using $\mathcal{R}$ to refer both to the expected reward function and the deterministic reward function. The general form of a stochastic reward function is: $R \sim \mathcal{R}(.|s,a,s')$. I suggest using $R_t$ to refer to samples from $\mathcal{R}(.|s,a,s')$ in your equations, such as Eq 1.


**Strength And Weaknesses:**

**Strengths:**

- The paper is very well written.

- The theoretical results are followed immediately by intuitive conclusions, making the paper very easy to follow.

- The arguments are made very systematically, with great flow.

- The way in which they attack their questions from a theoretical viewpoint is very clever.

- The experiments are nicely executed, and the results are reliable in my view.

**Weaknesses:**

- The argument that results don't translate monotonically from low-data regime to high-data regime or vice versa is effectively based on the assumption that distributional methods represent the highest-performing methods in high-data regime. This however is not really true in my view, as there currently are (and I'm sure there will be many more) non-distributional RL methods that are on par with or outperform distributional methods (e.g., if I remember correctly, Munchausen DQN [1] and Linear-Logarithmic DQN [2] methods are some such instances). As such, I feel the paper does not tell us much concretely on whether a general argument regarding Objective 2 (i.e. question of what is the relationship between sample complexity in different data regimes).

**References:**

[1] Nino Vieillard, Olivier Pietquin, Matthieu Geist (2020) *Munchausen Reinforcement Learning*. NeurIPS (Spotlight).

[2] Mehdi Fatemi, Arash Tavakoli (2022) *Orchestrated Value Mapping for Reinforcement Learning*. ICLR.



**Summary Of The Paper:**

The paper explores both theoretically and empirically the following questions:

1. Do the performance profiles of deep RL algorithms designed for certain data regimes (high-data regime) maintain monotonicity when tried in a different data regime (e.g. low-data regime)?

2. What is the underlying theoretical relationship between the performance profiles and sample complexity regimes?

Summary of theoretical results (in linear function approximation):
- The gap between performance in the low-data regime and the high-data/asymptotic regime can be *arbitrarily large*. Thus, comparisons between algorithms in the asymptotic/high-data regime are not informative when trying to understand algorithm performance in the low-data regime.

- Proves that distributional RL (with known finite support) has an intrinsically higher sample complexity than that of standard Q-learning.

- Proves that the sample complexity of distributional RL (with unknown support) can be significantly worse than that with known support.

The empirical study extensively that the performance of basic RL baselines (dueling DQN and Double DQN) are significantly better in low-data regimes than the SOTA methods in large-data regimes (various distributional RL methods). Thus, establishing the non-monotonic relationship between the performance in one data regime to another. Experiments are based on ALE 100k vs. 200M.

**Summary Of The Review:**

Please see my comment in the previous section on **Clarity, Quality, Novelty and Novelty**, which leads to my advocating for accepting this paper.

---

> ### Author Response · Authors · 2022-11-06
> **Author Response**
>
> Thank you very much for your insightful response. Below we try to address your questions.
>
> We completely agree with you. We have tried to uncover the implicit assumptions that are currently shaping the algorithm design in the low-data region, thus we have focused on algorithms that have been designed in the high-data regime that have been adopted in the low-data regime via researchers. The reason that some part of our study is focused on distributional reinforcement learning is simply because the instances of these implicit assumptions that exist in the low-data regime algorithm design studies contain comparisons with the Rainbow algorithm which builds upon distributional reinforcement learning. We were not able to find instances of Munchausen DQN and Linear-Logarithmic DQN being adopted in the low-data regime by other researchers. However, thank you for mentioning this, in case in the future when the researchers want to transfer these algorithms to the low-data regime, we hope that our paper might provide insights into the relationship between asymptotic sample complexity and low-data regime sample complexity.
>
> 1. *"Section 4 and linear FA case"*
>
> Thank you for this question. The results of Section 4 do not assume that the state-action values are given by linear function approximation. This section is dedicated to providing a high-level mathematical justification for why one should expect distributional reinforcement learning methods to require more samples than standard Q-learning methods.

---

> > ### Comment · Reviewer_88qX · 2022-11-16
> > **Thanks for your response**
> >
> > Thanks for your response which clarifies my questions.

---

### Official Review · Reviewer_kZsD · 2022-10-24

**Confidence:** 4
**Correctness:** 2
**Technical Novelty And Significance:** 2
**Empirical Novelty And Significance:** 3
**Recommendation:** 3

**Clarity, Quality, Novelty And Reproducibility:**

- The paper is overall well written and easy to follow
- The distributional reinforcement learning is not quite clear to me. Many notations like $\mathcal Z, v_\max, v_\min$ are not well explained.
- I did not find the code for the reproducibility check.
- The novelty issue is discussed above

**Strength And Weaknesses:**

### Strength:

- The paper is well-written and easy to follow. Extensive experiments well support the claimed result.
- Analysing the relationship between the performance profiles and sample complexity sounds interesting to me.

### Weakness:

- The word `Neural` in the title is not well supported in the paper. The results in Section 3 are built on a linear approximation following (Zanette et al. 2020). Sections 4 and 5 are regarding the sample complexity based on the concentration. There's no neural network analysis in this paper.
- The novelty of Section 3, especially Proposition 3.2 is not significant. It is just some direct result of applying some basic algebra given (Zanette et al. 2020) for linear function approximation.
- The contribution of Section 4 is not enough. It suggests that knowing the mean value is not sufficient for generating a good policy. In addition, learning the distribution would be more expensive compared with learning the mean value. I don't think this result is a new result given the existing RL literature.

**Summary Of The Paper:**

This paper studies the performance of the RL algorithms within the low data regime. Extensive experiments are carried out to demonstrate how well the performance profiles influence the sample complexity.

**Summary Of The Review:**

I have no more comments besides the comments above. Given the concern about the contribution and novelty of this paper, I don't think this paper meets the ICLR's bar.

---

> ### Author Response · Authors · 2022-11-06
> **Author Response**
>
> Thank you for investing your time to read our paper. Below we try to address your questions.
>
> The main drive of our paper is to realize the implicit assumptions of the monotonic translation of deep reinforcement learning algorithms designed for certain data regimes to different sample complexity regions. These implicit assumptions that currently exist in many recent studies (DRQ [ICLR 2021 Spotlight Presentation], DER [NeurIPS 2019], SimPLe [ICLR 2020 Spotlight Presentation], OTR, and Efficient Zero [NeurIPS 2021]) result in distributing faulty signals on what makes these algorithms work.
>
> Targeting these implicit assumptions, Section 6 is dedicated to thorough investigation of these instances and how these assumptions hurt how we evaluate algorithms as they move through regions, and how carrying these assumptions leads to comparisons that give faulty signals on why certain algorithms work or don’t work. These kinds of inaccurate signals on algorithm comparisons might cause certain biases in algorithm design, and hence shape future research directions and efforts to divert from promising algorithms. While Section 6 thoroughly demonstrates that the current comparison methodologies indeed cause certain biases in algorithm design, Section 3 and Section 4 are dedicated to providing more theoretical insights and justifications into the observations and results provided in Section 6.
>
> We would like to highlight that we do not posit our paper as a pure theory paper with deep theoretical contributions to the field, but rather propose theoretical justifications to explain the observations and results we have obtained in the main part of the paper.
>
> 1. *"Section 4"*
>
> While you claim that the expense of learning the distribution compared to learning the mean is clear, all of the low-data regime algorithm design papers (DRQ [ICLR 2021 Spotlight Presentation], DER [NeurIPS 2019], SimPLe [ICLR 2020 Spotlight Presentation], OTR, and Efficient Zero [NeurIPS 2021]) compare their proposed algorithms to distributional reinforcement learning algorithms. Thus resulting in faulty signals on the comparisons that are being made. Hence, we believe explicitly explaining these implicit assumptions being made in low-data regime algorithm design can provide an insightful view on the reasoning and the outcomes of these assumptions.
>
> Thank you again for investing your time for reviewing our paper. Please let us know if we were able to address your questions.

---

### Official Review · Reviewer_CAyy · 2022-10-25

**Confidence:** 4
**Correctness:** 4
**Technical Novelty And Significance:** 1
**Empirical Novelty And Significance:** 2
**Recommendation:** 3

**Clarity, Quality, Novelty And Reproducibility:**

The paper is reasonably clear.

The quality of the experimental work is acceptable, but the theoretical results are not particularly surprising.

On the originality side, the paper tries to present itself as the first one recognizing a fundamental difference between algorithms configurations for asymptotic and data-efficient settings.

**Strength And Weaknesses:**

Strengths:
- The problem of detecting which algorithms are best depending of whether we are interested in data efficiency or asymptotic performance is very important;
- The empirical comparison between different algorithms seems quite thorough.

Weaknesses:
- The paper tries to position itself as the first one discovering that reinforcement learning algorithms created with the goal of having better asymptotic performance are not automatically the best one in terms of data efficiency. This is very misleading, since this is a well-know problem (see, e.g., DER [van Hasselt et al., 2019]), and also reduces the overall contribution of this paper.
- The theoretical results seem to be not very compelling. In particular, despite contextualized for linear function approximation, Proposition 3.2 appears to be trivially true for practically any type of reinforcement learning algorithm. Likewise, the idea that learning a distribution potentially requires more samples compared to an expected value is common knowledge; more interesting and relevant is the interaction of such a learning process with representation learning, which has been shown to be one of the crucial aspects behind the success distributional reinforcement learning and is no object in the analysis presented in the paper.

**Summary Of The Paper:**

The paper analyzes, theoretically and empirically, the sample complexity of DQN-based algorithm on the ALE benchmark. It shows that the algorithms considered better in terms of asymptotic performance are actually worse than simpler baselines in the Atari 100k benchmark for sample efficiency.

**Summary Of The Review:**

In short, while I believe the study of the behavior of existing algorithms in the data-efficient and asymptotic setting to be an important avenue of research, I find both the theoretical analysis and the position of the current version of the paper to be particularly misleading and I thus recommend rejection.

---

> ### Author Response · Authors · 2022-11-06
> **Author Response**
>
> Thank you for the time you have invested in reading our paper and providing feedback.
>
> 1. *“DER [van Hasselt et al., 2019]”*
>
> Thank you for mentioning the DER paper. DER is currently cited in our paper and explained in detail in Section 6 and DER results are reported in Figure 4. One important thing we would like to highlight here is that the main contribution of the DER paper is to show the intrinsic commonality between parametric models and experience replay via providing a like-for-like comparison of Rainbow to parametric models (i.e. model based reinforcement learning).
>
> In particular, DER shows that the SimPLE algorithm which conducts comparisons against the Rainbow algorithm does not adjust the hyperparameters to test the algorithm in the low data-regime. Thus, DER as a response to this study thoroughly demonstrates a proper investigation towards the importance of experience replay and its adjustment to the low-data regime. Hence, as a result DER shows that with the proper adjustments DER performs better than SimPLE.
>
> On the other hand, our paper is focusing on the implicit assumption on the approximately monotonic translation of deep reinforcement learning algorithms designed for certain data regimes to different sample complexity regions. We highlight that these implicit assumptions are currently present in all of the recent studies that proposed algorithms in the low-data regime (e.g. DRQ [ICLR 2021 Spotlight Presentation], DER [NeurIPS 2019], SimPLe [ICLR 2020 Spotlight Presentation], OTR, and Efficient Zero [NeurIPS 2021]). The results of our paper demonstrate that this implicit assumption is incorrect, and results in conveying faulty signals on how and what makes these algorithms work or not work, and has led to inaccurate conclusions about the relative performance of recently proposed algorithms in the low data regime.
>
> Furthermore, we would like to highlight that, from direct communication with the authors of the DER paper, the DER authors find the results of our paper surprising, intriguing and an explicitly valuable contribution to the field.
>
>
> 2. *“Fundamental difference between algorithm configurations for asymptotic and data-efficient settings.”*
>
> We believe that you might have some confusion here. Our paper is not about the fundamental difference between algorithm **configurations** for asymptotic and data-efficient settings. Our paper is about the implicit assumptions on the approximately monotonic translation of deep reinforcement learning algorithms designed for certain data regimes to different sample complexity regions. As has been explained in Section 6, even with the right configuration of the Rainbow algorithm (i.e. DER) in the low data regime, DER does not perform at the same level with the baseline dueling algorithm. Note that DER also carries this implicit assumption in that Rainbow is the best performing algorithm in the high-data regime, thus it must be the best performing algorithm in the low-data regime. However, we show that the fact that the baseline dueling algorithm performs significantly better than DER clearly demonstrates the non-monotonicity of the performance profiles moving between sample complexity regions.
> While our paper thoroughly investigates the reasoning behind the instances of these implicit assumptions, it further provides the implications and outcomes of these comparison instances that currently exist in low-data regime algorithm design.
>
> 3. *"Proposition 3.2...true for practically any type of reinforcement learning algorithm"*
>
> We think you may have some confusion here about the implications of Proposition 3.2. In particular, Proposition 3.2 makes a statement about the regret achievable by the **optimal** algorithm, which will not be true in general for **any** algorithm.
>
> In particular the result of Zanette et al. shows that the **optimal** achievable regret for linear function approximation exhibits a particular quantitative trade-off in terms of the representation dimension and accuracy of function approximation. Proposition 3.2 then shows that, depending on the representation dimension and the approximation accuracy, this particular quantitative trade-off in the **optimal** achievable regret can lead to arbitrarily large performance differences between the high and low-data regime.
>
>
>
> We highly appreciate the time you have invested in reviewing our paper. We hope that we were able to clarify your questions. Please let us know if you have any further questions.

---

> ### Author Response · Authors · 2022-11-08
> **Discussion Period**
>
> We were hoping to have a vibrant discussion period. Can you please confirm that our response addresses your questions?

---

> ### Author Response · Authors · 2022-11-17
> **Discussion Period is Ending**
>
> The author response period will be ending in one day. Can you please confirm that our response addressed your questions? If you wanted to confer further we will be available to discuss.

---

> ### Author Response · Authors · 2022-11-25
> **Author Response Acknowledgement**
>
> Dear Reviewer CAyy,
>
> Thank you again for allotting your time to provide feedback on our paper. We responded to each of your questions in detail in the author response post below. We would highly appreciate it if you could read the responses, confirm that your questions have been addressed, and thus revise your review accordingly regarding the clarifications provided.
>
> Best regards,
>
> Authors

---

> ### Author Response · Authors · 2022-11-30
> **Reminder**
>
> Dear Reviewer CAyy,
>
> Thank you again for allotting your time to provide feedback on our paper. We responded to each of your questions in detail in the author response post below. We would highly appreciate it if you could read the responses, confirm that your questions have been addressed, and thus revise your review accordingly regarding the clarifications provided.
>
> Best regards,
>
> Authors

---

### Official Review · Reviewer_gGx6 · 2022-10-28

**Confidence:** 2
**Correctness:** 3
**Technical Novelty And Significance:** 2
**Empirical Novelty And Significance:** 3
**Recommendation:** 5

**Clarity, Quality, Novelty And Reproducibility:**

See above. The authors may wish to add pointers to implementations adopted (I guess they are directly adopted from the source papers) for better reproducibility.

**Strength And Weaknesses:**

$\textbf{Strength:}$

- The topic raised is interesting and important from both the theoretical and empirical perspectives. From the theoretical side, the common sense of "high sample efficiency" is mostly described by regret analysis, which aims to capture the long-term asymptotic performance of RL algorithms as the number of interaction rises. In contrast, in empirical studies, achieving "high sample efficiency" typically means beating SOTA on benchmarks with limited number of interactions. I am happy to see that the authors finally decide to look into the low-sample regime performance through the lens of recent theory analysis of sample efficient RL.

$\textbf{Weaknesses and Comments:}$

- The issue raised in this paper is more of a conceptual issue to me. To "gain superior performance with limited data/interaction" may not be accurately described as "achieving high sample efficiency" and does not indicate superior performance in rich data settings, but it is still an important research direction in the RL study.

- The discussion on distributional RL seems less related to the main point of this paper, though it is still related. The authors may wish to discuss more on how the performance of distributional RL is related to the point, namely, the relationship between performance profile and sample complexity regime.

- How may random seeds are utilized for the experiments?  Also, the experiments are not sufficient to me. It would be better if authors could give a more comprehensive survey on recent advances in exploration (e.g., RND, ICM ...)

- (Minor) The authors may wish to explicitly claim the implicit assumption behind exploration works (that is, algorithm performance transfers monotonically between the low-sample regime and high-sample regime, if I am on the right track of this paper) in the second paragraph of the introduction.







**Summary Of The Paper:**

 This paper aims to answer two questions, namely,

(i) Do the performance profiles of deep
reinforcement learning algorithms designed for certain data regimes translate approximately linearly to a different sample complexity region?

(ii) What is the underlying theoretical relationship between the performance profiles and sample complexity regimes?

To answer the first question, the authors conducted analysis and experiments on distributional RL algorithms on different sample complexity regimes and discover that the relationship is not monotonic (namely, superior performance in one sample complexity regime does not indicate superior performance in the other sample complexity regime). To answer the second question, the authors proved that the gap between low-data and high-data regime performance could be arbitrarily large in the linear MDP case.

**Summary Of The Review:**

The topic is interesting and important, but the paper might need further preparation for publication, in my opinion.

---

> ### Author Response · Authors · 2022-11-06
> **Author Response**
>
> Thank you for the time you have allocated in reading our paper, and providing feedback.
>
> 1. *“The discussion on distributional RL“*
>
> The reason that distributional RL is a part of the comparison benchmark is simply due to the fact that the Rainbow algorithm had been acknowledged as the best performing algorithm when the low-data regime focused papers were conducting all of their comparisons against Rainbow which utilizes distributional reinforcement learning. Thus, when we were investigating the sample complexity from the asymptotic to the low-data regime we wanted to include major algorithms that have been both designed in the high-data regime, and have been the subject of attempts for adoption in the low-data regime. Thank you for raising your suggestions on this. We can absolutely add more explanations on why distributional RL is part of this comparison.
>
> 2. *"How many random seeds are utilized for the experiments?"*
>
> 10 random seeds have been used in the experiments.
>
> 3. *“ICM and RND”*
>
> Perhaps our response to “the discussion on distributional RL” is also relevant to this point. Our main focus was to target instances of comparisons to algorithms that have been designed in the high-data regime and later have been further adopted in the low data regime by researchers. The instances of these, as also has been reported in Section 6, are DRQ [ICLR 2021 Spotlight Presentation], DER [NeurIPS 2019], SimPLe [ICLR 2020 Spotlight Presentation], OTR, and Efficient Zero [NeurIPS 2021]. On the other hand, to the best of our knowledge we could not find any effort on the ICM [1] and RND [2] (please let us know if we correctly identified the papers you have mentioned) to be adopted in the low data regime.
>
> [1] Yuri Burda, Harrison Edwards, Amos Storkey, Oleg Klimov. Exploration by Random Network Distillation, ICLR 2019.
>
> [2] Deepak Pathak, Pulkit Agrawal, Alexei A. Efros, Trevor Darrell. Curiosity-driven Exploration by Self-supervised Prediction, ICML 2017.

---

> ### Author Response · Authors · 2022-11-29
> **Kind Reminder**
>
> Dear Reviewer gGx6,
>
> Thank you again for allocating your time to provide feedback on our paper. Would it be possible for you to read our responses and confirm that your questions have been addressed?
>
> Best regards,
>
> Authors

---

### Decision · Program_Chairs · 2023-01-20

**Decision:**

Reject

**Justification For Why Not Higher Score:**

The metareviewer believes the paper is not ready for publication. In addition to the valid concerns pointed out by the reviewers, the main theory in this paper does not seem convincing to prove their claim.

**Justification For Why Not Lower Score:**

N/A

**Metareview: Summary, Strengths And Weaknesses:**

Summary:
This paper studies the performance of deep RL algorithms in high-data and low-data regimes. The authors conducted analysis and experiments on distributional RL algorithms on different data-size regimes and discover that superior performance in one data-size does not indicate superior performance in the other regime. The authors proved that the gap between low-data and high-data regime performance could be arbitrarily large in the linear MDP case.

Strength:
- The topic is interesting and important from both the theoretical and empirical perspectives.
- The paper is well-written and easy to follow.
- The empirical comparison between different algorithms seems thorough.

Weakness:
- The discussion on distributional RL seems less related to the main point.
- The theoretical results seem to be not very compelling. The metareviewer also believes that theory is misaligned with the main argument in the paper: for instance, in Prop. 3.2, the lower bound is for "any algorithms," and the upper bound is for the "good algorithm" (of the high-data regime). It does not say anything about why the "good algorithm" in the low-data regime cannot beat other algorithms (e.g., a baseline algorithm).


**Summary Of Ac-Reviewer Meeting:**

While other reviewers did not respond, the reviewer holding the positive comments believes the paper should be accepted by its merit in large scale experiments. Nevertheless, the metareviewer believes the paper is not ready for publication. In addition to the valid concerns pointed out by the reviewers, the main theory in this paper does not seem convincing to prove their claim.